

# Luminescent characteristics and mitochondrial COI barcodes of nine cohabitated Taiwanese fireflies

King-Siang Goh[1], Liang-Jong Wang[2], Jing-Han Ni[3] and Tzi-Yuan Wang[4]

[1] Genomics Research Center, Academia Sinica, Taipei, Taiwan
[2] Forest Protection Division, Taiwan Forestry Research Institute, Taipei, Taiwan
[3] Department of Ecological Humanities, Providence University, Taichung, Taiwan
[4] Biodiversity Research Center, Academia Sinica, Taipei, Taiwan

## ABSTRACT

**Background**. Over 50 Taiwanese firefly species have been discovered, but scientists lack information regarding most of their genetics, bioluminescent features, and cohabitating phenomena. In this study, we focus on morphological species identification and phylogeny reconstructed by *COI* barcoding, as well as luminescent characteristics of cohabited Taiwanese firefly species to determine the key factors that influenced how distinct bioluminescent species evolved to coexist and proliferate within the same habitat.

**Methods**. In this study, 366 specimens from nine species were collected in northern Taiwan from April to August, 2016–2019. First, the species and sex of the specimens were morphologically and genetically identified. Then, their luminescent spectra and intensities were recorded using a spectrometer and a power meter, respectively. The habitat temperature, relative humidity, and environmental light intensity were also measured. The cytochrome oxidase I (COI) gene sequence was used as a DNA barcode to reveal the phylogenetic relationships of cohabitated species.

**Results**. Nine species—eight adult species (*Abscondita chinensis, Abscondita cerata, Aquatica ficta, Luciola curtithorax, Luciola kagiana, Luciola filiformis, Curtos sauteri,* and *Curtos costipennis*) and one larval *Pyrocoelia praetexta*—were morphologically identified. The nine species could be found in April–August. Six of the eight adult species shared an overlap occurrence period in May. Luminescent spectra analysis revealed that the $\lambda_{max}$ of studied species ranged from 552–572 nm (yellow–green to orange–yellow). The average luminescent intensity range of these species was about 1.2–14 lux (182.1–2,048 nW/cm$^2$) for males and 0.8–5.8 lux (122.8–850 nW/cm$^2$) for females, and the maximum luminescent intensity of males was 1.01–7.26-fold higher than that of females. Compared with previous studies, this study demonstrates that different $\lambda_{max}$, species-specific flash patterns, microhabitat choices, nocturnal activity time, and/or an isolated mating season are key factors that may lead to the species-specific courtship of cohabitated fireflies. Moreover, we estimated that the fireflies start flashing or flying when the environmental light intensity decreased to 6.49–28.1 lux. Thus, based on a rough theoretical calculation, the sensing distance between male and female fireflies might be 1.8–2.7 m apart in the dark. In addition, the mitochondrial COI barcode identified species with high resolution and suggested that most of the studied species have been placed correctly with congeners in previous phylogenies. Several cryptic species were revealed by the COI barcode with 3.27%–12.3% variation.

Corresponding author
Tzi-Yuan Wang, tziyuan@gmail.com

This study renews the idea that fireflies' luminescence color originated from the green color of a Lampyridae ancestor, then red-shifted to yellow-green in Luciolinae, and further changed to orange–yellow color in some derived species.

## INTRODUCTION

Among terrestrial bioluminescent insects, fireflies (Lampyridae) have the most charismatic shine, which they use for mating or aposematic signals at night (*Oba, Branham & Fukatsu, 2011*). Fireflies in Coleoptera are the most diverse terrestrial group of bioluminescent organisms. Over 2,100 firefly species have been reported in temperate and tropical regions, including Eurasia, America, New Zealand, and Australia. Firefly life history and bioluminescence have been studied for over a century and have offered bioinspiration for many inventions and methods, such as a method for detecting gene expression (biomedical), improvements in LED technology (industrial), and algorithms (mathematical) (*Kaskova, Tsarkova & Yampolsky, 2016*; *Kim et al., 2016*; *Yang, 2009*). Fireflies are also considered to be an environmental indicator species for assessing light, water, and soil pollution. Moreover, some of their larvae—such as *Pyrocoelia pectoralis*, which eat invasive snails (*Fu & Meyer-Rochow, 2013*)—are used as biological controls in some species. Firefly population sizes are dramatically affected by changes in land-use, as habitat deterioration and artificial night lighting decrease their populations (*Firebaugh & Haynes, 2016*; *Owens, Meyer-Rochow & Yang, 2018*).

The phylogeny of Lampyridae (fireflies) has been reassessed several times (*Ballantyne et al., 2013*; *Ballantyne & Lambkin, 2013*; *Ballantyne et al., 2019*; *Chen et al., 2019*; *Martin et al., 2017*; *Martin et al., 2015*; *Martin et al., 2019*; *Stanger-Hall, Lloyd & Hillis, 2007*; *Wang, Wu & Wang, 2021*). These studies identified the following subfamilies: Ototretinae, Cyphonocerinae, Luciolinae (incl. *Pristolycus*), Pterotinae, Lamprohizinae, Psiocladinae, Amydetinae, Photurinae, and Lampyrinae. The most comprehensive study used 436 genomic loci to reconstruct a consensus phylogeny of fireflies with paraphyletic subfamilies, except Ototretinae with *Drilaster* and *Stenocladius* (*Martin et al., 2019*). For example, this study reassessed Luciolinae as paraphyletic with *Lamprigera*, and the higher-level classification of Lampyridae was revised accordingly. However, only few Asian species were included. In addition, the reassessed phylogeny might influence the hypothesis of previous bioluminescent evolution (*Martin et al., 2017*; *Oba et al., 2020*).

Previous phylogeny of fireflies reveals the evolution of their bioluminescence (*Martin et al., 2017*; *Oba et al., 2020*). Studies show that luminescence appeared in the common ancestor of Lampyridae about 100–200 million years ago (*Oba et al., 2020*; *Zhang et al., 2020*). In the forests of the mid-Cretaceous, the first luciferase gene evolved from acyl-CoA synthetase (acyl-CoA synthetase) to produce yellow luminescence that may be due to nocturnal predation. The ancestral Lampyrinae fireflies later evolved to have green

luminescence, while the ancestral Luciolinae fireflies evolved a red-shifted yellow-green luminescence; more species need to be studied to confirm this evolutionary distinction.

Most fireflies glow during the larval stage (1–2 years), but bioluminescent courtship behavior only occurs during the short adult stage (2–4 weeks) (*Buck, 1948*; *Riley, Rosa & Lima da Silveira, 2021*). All known luminous signals of adult fireflies can be roughly divided into flashing and continuous glowing (*Lloyd, 1966*; *Seliger et al., 1964*). Research suggests that each species has its own specific flash pattern, determined by differences in flash duration, flash frequency, and flash color (*Lewis & Cratsley, 2008*; *Lloyd, 1966*; *Seliger et al., 1964*). The wavelength ($\lambda_{max}$) of most fireflies' flash color range from yellow-green (538 nm) to orange-red (622 nm) (*He et al., 2021*).

The mitochondrial *Cox1* (*COI*) barcode is a powerful biomarker for estimating large-scale species richness, determining the potential for beta-diversity studies, and setting conservation priorities. However, error rates can be high for some individual genera, especially when very recent species form nonmonophyletic clusters (*Bergsten et al., 2012*; *Hendrich et al., 2015*; *Pentinsaari et al., 2016*). The comprehensive *COI* barcode databases and the Barcode Index Number (BIN) system are well-established and regularly updated (*Adamowicz, 2015*; *Adamowicz et al., 2017*; *Hendrich et al., 2015*; *Ratnasingham & Hebert, 2013*; *Roslin et al., 2022*; *Rulik et al., 2017*). In insects such as beetles, the mitochondrial *COI* barcode has proven an effective molecular marker for species identification (*Hendrich et al., 2015*; *Pentinsaari, Hebert & Mutanen, 2014*; *Raupach et al., 2020*; *Roslin et al., 2022*). The *COI* barcode can also be used to establish firefly phylogeny, biogeography, and population genetics, as well as to identify cryptic species (*Choi et al., 2003*; *Dong et al., 2021*; *Han et al., 2020*; *Jusoh et al., 2014*; *Kim et al., 2001*; *Lee et al., 2003*; *Muraji, Arakaki & Tanizaki, 2012*; *Stanger-Hall, Lloyd & Hillis, 2007*; *Usener & Cognato, 2005*). Thus, the *COI* barcode is a cheaper and more convenient biomarker for firefly identification. However, until the past two years, only a few Asian species had been sequenced (*Choi et al., 2003*; *Dong et al., 2021*; *Han et al., 2020*; *Kim et al., 2001*; *Lee et al., 2003*; *Liu & Fu, 2020*; *Sriboonlert & Wonnapinij, 2019*).

Fifty-six species have been described from Taiwan to date (*Jeng, Lai & Yang, 2003*; *Jeng, Yang & Engel, 2007*; *Jeng et al., 1998*), but few reports have been made on their biodiversity, ecological habitats, comparative morphology, life cycle, or behavior (*Ballantyne et al., 2013*; *Ballantyne et al., 2015*; *Ballantyne & Lambkin, 2013*; *Ballantyne et al., 2019*; *Goh, Lee & Wang, 2022*; *Goh & Li, 2011*; *Ho et al., 2010*; *South et al., 2008*). Taiwan has *Luciola*; *Curtos*; the reassessed *Aquatica* (*Fu, Ballantyne & Lambkin, 2010*), *Abscondita* (*Ballantyne et al., 2013*; *Ballantyne et al., 2019*), and *Sclerotia* (*Ballantyne et al., 2016*) species of Luciolinae; and some Lampyrinae species (*Ballantyne et al., 2013*; *Goh, Lee & Wang, 2022*; *Jeng, Lai & Yang, 1999*; *Jeng, Yang & Engel, 2007*; *Jeng et al., 1998*; *Ohba & Yang, 2003*; *Wang, Wu & Wang, 2021*). The endemic *Abscondita cerata* (formerly known as *Luciola cerata*) is the most abundant species, widely distributed from low altitude to medium-high altitude (1,500 m) in Taiwan. During its breeding season, several sympatric fireflies could be found (*Goh, Lee & Wang, 2022*; *Jeng, Lai & Yang, 1999*; *Ohba & Yang, 2003*). A recent study revealed that LED light intensity can influence the flash pattern of *Aquatica ficta* (*Owens, Meyer-Rochow & Yang, 2018*). These are the only two species of Taiwanese firefly for which systematic

**Table 1** Luminescent spectrum ($\lambda_{max}$) and intensity (nW/cm$^2$) of nine cohabitated species from two habitats in northern Taiwan.

| Species | Sex | Individuals ($n$) | $\lambda_{max}$ (nm) | Luminescent intensity [b],[c] (nW/cm$^2$) | |
| --- | --- | --- | --- | --- | --- |
| | | | | Mean | Maximum |
| *Abscondita cerata* | Female | 17 | 562.2 ± 0.4 | 122.8 ± 19.3 | 282.2 |
| | Male | 28 | 563.6 ± 0.3 | 406.6 ± 96.5 | 2,048 |
| *Abscondita chinensis* | Female | 3 | 571.3 ± 0.3 | 245.7 ± 83.9 | 329.7 |
| | Male | 2 | 572.0 ± 0.0 | 332.1 | 332.1 |
| *Aquatica ficta* | Female | 5 | 564.0 ± 0.5 | 569.4 ± 101.1 | 850 |
| | Male | 17 | 564.4 ± 0.3 | 525.7 ± 71.1 | 1,102 |
| *Curtos costipennis* | Female | 1 | 554 | 462 | ND |
| | Male | – | – | – | – |
| *Curtos sauteri* | Female | 5 | 554.0 ± 0.3 | 187.7 ± 55.7 | 349.3 |
| | Male | 3 | 552.7 ± 0.9 | 347.3 ± 95.9 | 536.7 |
| *Luciola curtithorax* | Female | 12 | 566.3 ± 0.4 | 157.9 ± 30.4 | 301.3 |
| | Male | 26 | 572.5 ± 0.2 | 356.1 ± 48.0 | 814.1 |
| *Luciola filiformis* | Female | – | – | – | – |
| | Male | 12 | 567.3 ± 0.2 | 182.1 ± 31.2 | 323.8 |
| *Luciola kagiana* | Female | 3 | 574.3 ± 0.3 | ND | ND |
| | Male | 3 | 574.0 ± 1.0 | 5.4 ± 4.8 | 10.2 |
| *Pyrocoelia praetexta* | Larva[a] | 3 | 552.7 ± 0.9 | ND | ND |

**Notes.**
[a] Luminescent spectra were only successfully recorded from larvae.
[b] Mean and maximum values were obtained as described in Methods and materials.
[c] "–": no sample; "ND": not detectable.

studies have been conducted based on luminescence spectrum and DNA barcoding. Therefore, this study investigated nine cohabitated species in northern Taiwan for species identification by *COI* barcode, flash color and luminescent intensity to determine the key factors through which distinct bioluminescent species evolved to coexist and proliferate within the same habitat.

# MATERIALS & METHODS

## Specimen collection and habitat

366 specimens of eight adult and one larval species were randomly collected in flight or from vegetation using hand dip nets from two habitats in the suburbs of Taipei, Taiwan— Nankang (25°01′40.4″N 121°38′02.6″E) and Miaoli County, Nanzhuang (24°37′53.5″N 121°01′37.0″E)—at 18:30–19:30 from April to August 2016–2019 (Table 1). After the bioluminescent spectrum/intensity measurement, the specimens were deposited in a laboratory freezer. For DNA extraction, several specimens were then stored at −80 °C. The remaining specimens were stored at −20 °C for species identification.

The environmental conditions before and after the fireflies began flashing and/or flying were investigated to understand what environmental factors may trigger their nocturnal activity. The temperature, relative humidity, and light intensity (lux) of the firefly habitat

were recorded during the period before flashing/flying (18:20~18:40) and the period after the fireflies began flashing/flying (18:30~18:50) using HOBO U12-012 data loggers (Onset Computer Corp., Bourne, MA, USA) at 10-sec intervals.

## Species identification and morphological measurements

The specimens were collected as previously described in *Goh, Lee & Wang (2022)*. The material collected in this study was identified by LJ Wang on the species level through the use of available references (*Ballantyne et al., 2013*; *Chen, 2003*; *Jeng, Lai & Yang, 2003*; *Jeng, Yang & Lai, 2003*; *Jeng et al., 1998*). Five morphological characteristics of the specimens (body length, pronotum length, pronotum width, front wing length, and front wing width) were measured (see Table S3) using a dissecting microscope and photographed with a digital video camera as previously described in *Goh, Lee & Wang (2022)*. During the survey, the specimens were chilled on ice. All surveys were completed within two days of the collection. One to five identified specimens were sacrificed and stored at −20 °C in the Biodiversity Research Center, Academia Sinica, Taipei, Taiwan (contact person: TY Wang, tziyuan@gmail.com).

## Bioluminescence spectrum/intensity measurement

The wavelength ($\lambda_{max}$) and luminescent intensity ($nW/cm^2$) of the light flashes produced by the living samples were measured by a USB2000+ spectrometer (Ocean Optics) and a PD300 power meter (Ophir), respectively. All surveys were completed within two days of the collection. The wavelength and luminescent intensity measurements were performed in a dark room by directly attaching the detector of the USB2000+ spectrometer or PD300 to the light organ of a trapped firefly (Fig. 1 modified from *Goh, Lee & Wang (2022)*). The average wavelength peak and $\lambda_{max}$ were obtained from an average of 3–5 measurements in complete darkness at 25 °C with 75% humidity. The luminescent intensity of the flash was obtained by averaging each flash from 3–10 min of recording data with a PD300 power meter. To compare the luminescent intensity data from PD300 and HOBO U12-012 using the same units, all data in the energy unit $nW/cm^2$ were converted into lux *via* the conversion 1 lux = 1.464E−07 W/c $m^2$ = 146.41 $nW/cm^2$ (at 555 nm).

## Statistics

The differences in bioluminescence spectrum among the specimens were determined by the Chi-square test between two species.

## DNA barcode sequencing

Crude DNA was extracted from thoracic muscles *via* the ZR Tissue & Insect DNA MicroPrep[TM] kit (D6015). Two beetle-specific primers (ClepFolF 5′-ATTCAACCAATCATAAAGATATTGG-3′ and ClepFolR 5′-TAAACTTCTGGA TGTCCAAAAAATCA-3′) were designed based on the comprehensive DNA barcode database of beetles (*Hendrich et al., 2015*) to amplify a 620-bp segment including the cytochrome oxidase I (COI) gene. Polymerase chain reactions (PCRs) in 50-µL volumes were performed with a dNTP concentration of 200 µM and a primer concentration of 0.3 µM, with 50 ng of genomic DNA, one unit of TaKaRa Taq[TM] DNA Polymerase, and the
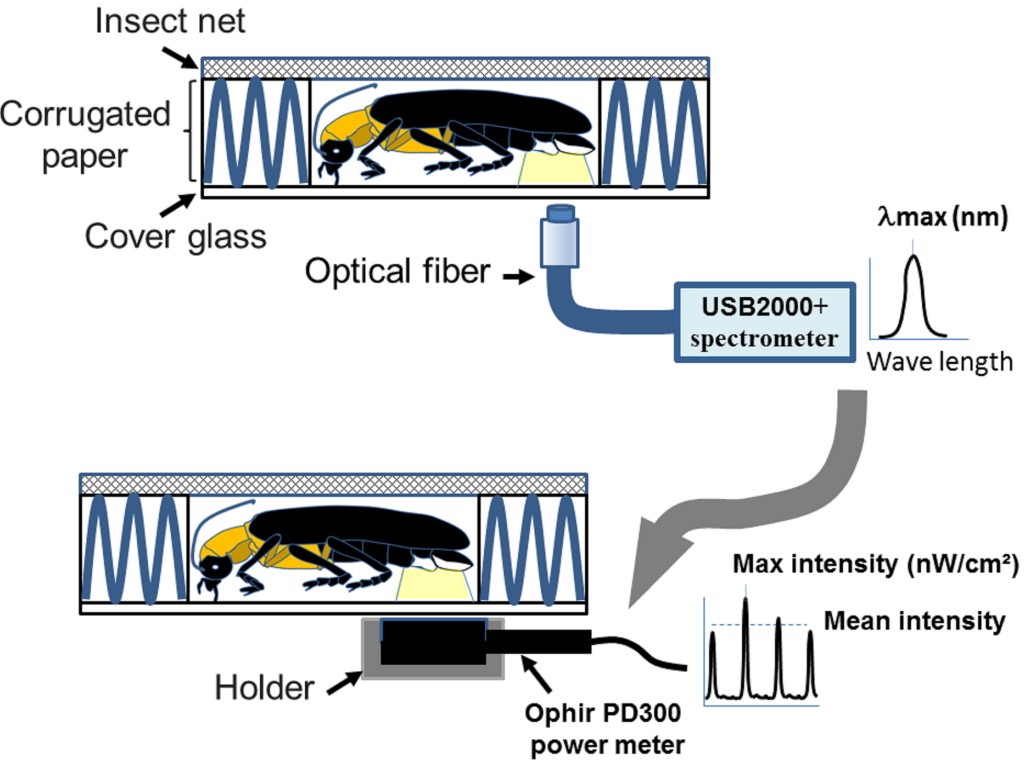

**Figure 1** **A homemade firefly holder for measuring flash wavelength and intensity adopted from *Goh, Lee & Wang (2022)*.** The measurements were performed by immobilizing individual fireflies in the hollow chamber ($1 \times 1.5 \times 0.4$ cm$^3$) of a homemade holder. The holder consists of an upper steel insect net (mesh pore size 0.1 cm), middle corrugated paper ($5 \times 5 \times 0.4$ cm$^3$), and a lower cover glass ($2.2 \times 2.2 \times 0.017$ cm$^3$). The coreless endcap of the optical fiber connecting to the spectrometry or power-meter is placed below the light organ for collecting the wavelength and intensity. Figure 1 was adapted from *Goh, Lee & Wang (2022)*.

buffer supplied by the manufacturer. The PCR was run for 35–40 cycles under the following conditions: denaturation at 95 °C for 30 s, annealing at 50~55 °C for 40 s, extension at 72 °C for 1 min, and a final extension at 72 °C for 10 min. The product mixture was used as a template for DNA sequencing (Genomics Ltd., Taipei, Taiwan). Haplotype sequences were deposited into GenBank under accession numbers MT534191–MT534201, ON209457 (Table 2).

## Molecular phylogeny

The COI sequences of closely-related species and/or species with known $\lambda_{max}$ of luminescence were downloaded from GenBank based on previous studies (*Adamowicz, 2015*; *Arnoldi, Neto & Viviani, 2010*; *Cassata, 2020*; *Dong et al., 2021*; *Han et al., 2020*; *He et al., 2021*; *Hendrich et al., 2015*; *Jusoh et al., 2021*; *Jusoh et al., 2018*; *Jusoh et al., 2014*; *Kim et al., 2001*; *Li, Yang & Fu, 2022*; *Liu et al., 2017*; *Liu & Fu, 2020*; *Martin et al., 2017*; *Martin et al., 2015*; *Muraji, Arakaki & Tanizaki, 2012*; *Oba, Branham & Fukatsu, 2011*; *Oba et al., 2020*; *Osozawa et al., 2015*; *Roslin et al., 2022*; *Rulik et al., 2017*; *Stanger-Hall, Lloyd & Hillis, 2007*; *Usener & Cognato, 2005*; *Wilcox, 2021*; *Zhang et al., 2018*). Sequences were
10.7717/peerj.14195
2022
Goh et al.

**Table 2   DNA barcodes (COI) of studied fireflies.** Only haplotype sequences were submitted to GenBank for the same species.

| Species | Accession number of haplotype (individual number) | Reference |
|---------|--------------------------------------------------|-----------|
| Luciolinae: | | |
| *Abscondita cerata* | MT534192 (6), MT534199 (3) | present study |
| *Abscondita chinensis* | MT534196 (3), ON209457 (1) | Present study |
| *Aquatica ficta* | MT534197 (2) | Present study |
| *Curtos sauteri* | MT534198 (1) | Present study |
| *Luciola curtithorax* | MT534191 (1), MT534193 (1), MT534195 (1) | Present study |
| *Luciola filiformis* | MT534201 (1) | Present study |
| *Luciola kagiana* | MT534200 (1) | Present study |
| Lampyrinae: | | |
| *Pyrocoelia praetexta* | MT534194 (1) | Present study |

then aligned using the ClustalX program (*Thompson, Gibson & Higgins, 2002*) with the default setting in MEGA X (*Kumar, Stecher & Tamura, 2016*), followed by length trimming due to different amplicons. After trimming, the short-length sequences were removed. At most, three representative sequences were kept for each species to simplify the tree topology. There were a total of 520 positions and 161 nucleotide sequences in the final dataset (Table S1). Neighbor-joining (NJ) (*Saitou & Nei, 1987*) and maximum-likelihood (ML) trees were constructed using GTR+G+I distances in MEGA X with 500 bootstrap replications (*Felsenstein, 1985*). The substitution model (parameter) used to calculate GTR+G+I distances (*Nei & Kumar, 2000*) was selected using Modeltest v3.7 (*Posada & Crandall, 1998*). The differences in the composition bias among sequences were considered in the evolutionary comparisons (*Tamura & Kumar, 2002*).

# RESULTS

## Cohabitated species composition at Nanzhuang and Nankang

From April to August 2016–2019, we collected 366 flying specimens from two firefly habitats (Nanzhuang and Nankang) in northern Taiwan (Table 1 & S2 Fig. 2), and morphologically identified them to the species level (Table S3). These specimens comprised nine different species, including adult males and/or females of *Aquatica ficta, Luciola filiformis, Abscondita cerata, Luciola kagiana, Luciola curtithorax, Abscondita chinensis, Curtos sauteri,* and *Curtos costipennis*. Only *Pyrocoelia praetexta* was observed as larvae from April to August. Five of the species—*Aq. ficta, L. filiformis, Abs. cerata, L. kagiana* and *P. praetexta*—were found in both habitats. *Luciola curtithorax* was collected only in Nankang, and *Abs. chinensis, C. sauteri,* and *C. costipennis* were collected only in Nanzhuang (Table S2). To simplify the results, Table 1 combined all specimens from the two habitats for further analysis.

## Occurrence periods of the cohabitated species

Figure 3 shows the estimated occurrence periods of the eight adult species in Nankang and Nanzhuang based on the collection dates of the specimens. *Abscondita cerata* and *L. kagiana* occurred in April–May, while *C. sauteri* and *Aq. ficta* occurred in May–August.

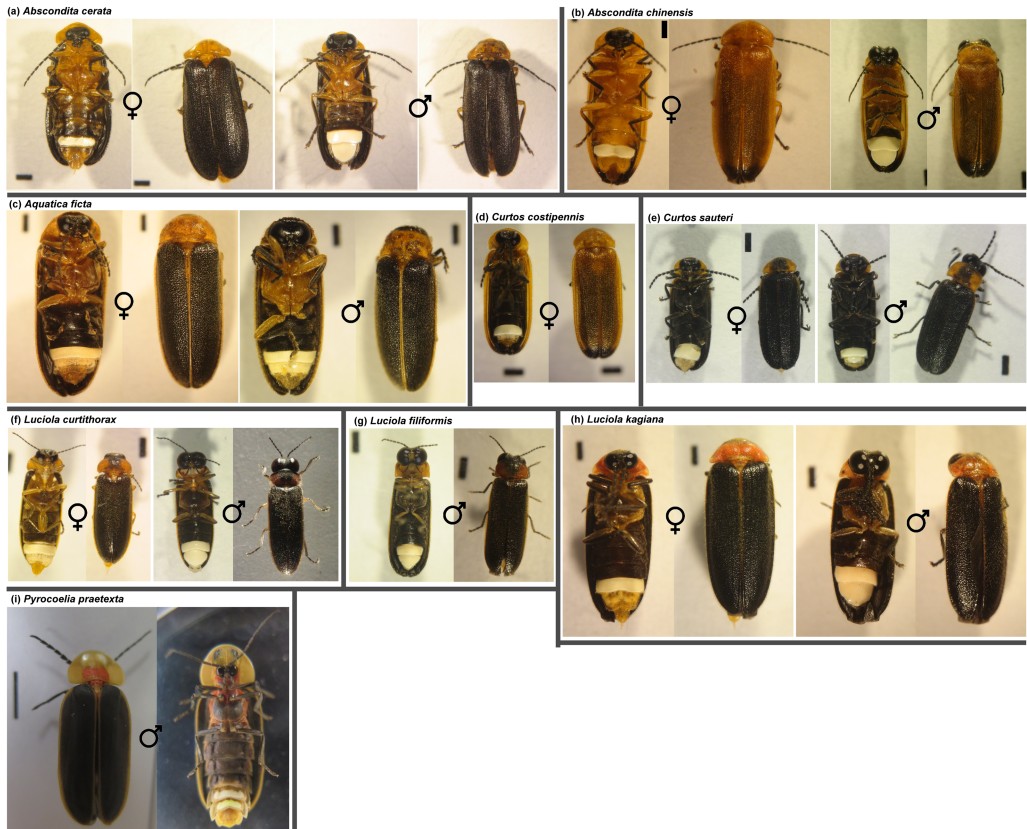

**Figure 2** **Representative females and males of collected firefly species.** (A) *Abscondita cerata* (B) *Abscondita chinensis* (C) *Aquatica ficta* (D) *Curtos costipennis* (E) *Curtos sauteri* (F) *Luciola curtithorax* (G) *Luciola filiformis* (H) *Luciola kagiana* (I) *Pyrocoelia praetexta*. The standard scale bar is one mm, except for *Pyrocoelia praetexta*, which uses a five mm scale bar.

*L. curtithorax* occurred in May-July while *L. filiformis* occurred in May–June. *Abscondita chinensis* was found only in June, while *C. costipennis* was found only in August. Six of the eight studied species shared an overlap occurrence period in May: *C. sauteri*, *Aq. ficta*, *Abs. cerata*, *L. kagiana*, *L. filiformis*, and *L. curtithorax*. The occurrence periods of this study overlapped with the occurrence periods of previous studies based in other habitats in Taiwan (*Chen, 2003*; *Chen & Jeng, 2012*).

## Differences in the luminescence spectrum between the cohabitated species

The average $\lambda_{max}$ from the luminescent spectra of nine studied species (eight adult species and larval *P. praetexta*) ranged from about 552 nm (green-yellow) to 575 nm (yellow-orange) (Table 1, Fig. S1). Excluding insufficient data on three species, the average $\lambda_{max}$ of five species showed no significant difference between intraspecific males and females. The average $\lambda_{max}$ of *L. curtithorax* was significant different between female and male. Ignoring the sexual differences, the pairwise comparison of interspecific $\lambda_{max}$ (Table 3) showed that the studied species commonly displayed significant difference ($p$-values <0.05) in $\lambda_{max}$ to
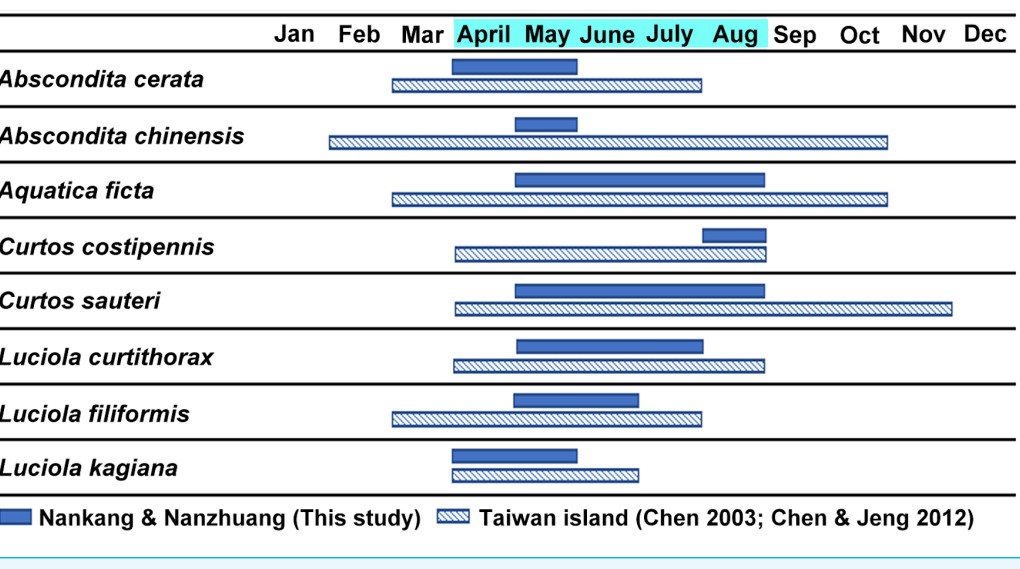

**Figure 3** The estimated occurrence periods of each adult species based on the collection dates of the specimens in the two studied sites (April–August) and previous literature (*Chen, 2003*; *Chen & Jeng, 2012*).

**Table 3** Differences in pairwise $\lambda_{max}$ (*p*-value) between species. The statistics were calculated using combined $\lambda_{max}$ from females and males. Numbers in boldface are not significantly different.

|  | *Abs. cerata* | *Abs. chinensis* | *Aq. ficta* | *C. sauteri* | *L. filiformis* | *L. curtithorax* | *L. kagiana* |
|---|---|---|---|---|---|---|---|
| *Abs. cerata* |  |  |  |  |  |  |  |
| *Abs. chinensis* | 0.0000 |  |  |  |  |  |  |
| *Aq. ficta* | 0.0012 | 0.0000 |  |  |  |  |  |
| *C. sauteri* | 0.0000 | 0.0000 | 0.0000 |  |  |  |  |
| *L. filiformis*[a] | 0.0000 | 0.0000 | 0.0000 | 0.0000 |  |  |  |
| *L. curtithorax* | 0.0000 | **0.0604** | 0.0000 | 0.0000 | 0.0000 |  |  |
| *L. kagiana* | 0.0000 | 0.0018 | 0.0000 | 0.0000 | 0.0000 | 0.0000 |  |
| *P. praetexta*[b] | 0.0000 | 0.0000 | 0.0000 | **0.1610** | 0.0000 | 0.0000 | 0.0000 |

**Notes.**
[a] Only adult male.
[b] Only larva could be detectable.

most other studied species. No significant difference was found between *Abs. chinensis* and *L. curtithorax* (*p*-value = 0.0604) or between *C. sauteri* and *P. praetexta* (*p*-value = 0.161).

To determine the courtship behaviors of cohabited fireflies based on sex, we further compared the $\lambda_{max}$ between six interspecific adult females (Table 4) and males (Table 5). Most studied species revealed significant differences in $\lambda_{max}$ between interspecific females. However, no significant difference has found between females of *Aq. ficta* and *Abs. cerata* (*p*-value = 0.0742). In contrast, the $\lambda_{max}$ comparison between interspecific adult males (Table 5) showed no significant difference between *Aq. ficta* and *Abs. cerata* (*p*-value = 0.672), between *L. curtithorax* and *Abs. chinensis* (*p*-value = 0.626), between *L. kagiana*

**Table 4 Differences in pairwise $\lambda_{max}$ ($p$-value) of adult female between species.** The statistics were calculated using $\lambda_{max}$ of adult females. Numbers in boldface are not significantly different.

|  | Abs. cerata | Abs. chinensis | Aq. ficta | C. sauteri | L. filiformis | L. curtithorax | L. kagiana |
|---|---|---|---|---|---|---|---|
| Abs. cerata |  |  |  |  |  |  |  |
| Abs. chinensis | 0.0000 |  |  |  |  |  |  |
| Aq. ficta | **0.0742** | 0.0000 |  |  |  |  |  |
| C. sauteri | 0.0000 | 0.0000 | 0.0000 |  |  |  |  |
| L. filiformis[a] | NA[b] | NA | NA | NA |  |  |  |
| L. curtithorax | 0.0000 | 0.0000 | 0.0095 | 0.0000 | NA |  |  |
| L. kagiana | 0.0000 | 0.0000 | 0.0000 | 0.0000 | NA | 0.0000 |  |

Notes.

[a]Without adult female.

[b]"NA": not analysis due to lack of female.

**Table 5 Differences in pairwise $\lambda_{max}$ ($p$-value) of adult males between species.** The statistics were calculated using $\lambda_{max}$ of males. Numbers in boldface are not significantly different.

|  | Abs. cerata | Abs. chinensis | Aq. ficta | C. sauteri | L. filiformis | L. curtithorax | L. kagiana |
|---|---|---|---|---|---|---|---|
| Abs. cerata |  |  |  |  |  |  |  |
| Abs. chinensis | 0.0000 |  |  |  |  |  |  |
| Aq. ficta | **0.0672** | 0.0000 |  |  |  |  |  |
| C. sauteri | 0.0028 | 0.0021 | 0.0031 |  |  |  |  |
| L. filiformis[a] | 0.0000 | 0.0000 | 0.0000 | 0.0022 |  |  |  |
| L. curtithorax | 0.0000 | **0.0626** | 0.0000 | 0.0010 | 0.0000 |  |  |
| L. kagiana | 0.0052 | **0.1835** | 0.0075 | 0.0001 | 0.0177 | **0.2606** |  |

and *Abs. chinensis* ($p$-value = 0.1835), and between *L. kagiana* and *L. curtithorax* ($p$-value = 0.2606).

## Correlation between firefly luminescent intensity and environmental photic intensity

This study was performed during an *Abs. cerata* massive occurrence (April to May) in Nankang and Nanzhuang. During the studied periods, it was estimated that the average environmental light intensity during twilight, the ten-minute period before the fireflies started flashing or flying in the habitats, was in a range of 35.7–136.5 lux (Table 6). The suitable environmental light intensity for fireflies flashing and/or flying was in a range of 6.49–28.1 lux.

With the exception of *L. kagiana* and *P. praetexta* due to abnormal behavior (no glowing or glowing in extremely low light intensity), the luminescent intensity of seven adult species was about 1.2–14 lux (182.1–2,048 nW/cm$^2$) in male fireflies and nearly 0.8–5.8 lux (122.8–850 nW/cm$^2$) in female fireflies (Table 1). The results showed that the male fireflies have higher luminescent intensity than the females, which might be related to their courtship behaviors.

Herein, we argue that firefly luminescent intensity is correlated with environmental photic intensity. For examples, among the studied male species, male *A. cerata* produced the brightest flashes, measuring up to 14 lux (or 2,048 nW/cm$^2$). In contrast, female *A.*

**Table 6   The environmental temperature, relative humidity, and environmental light intensity of the habitats around twilight and when *Abs. cerata* starts flashing/flying.**

| Date | Nocturnal activity time | Temp (°C) | RH (%) | Environmental light intensity (lux) |
|---|---|---|---|---|
| **A. Nankang, Taipei:** | | | | |
| 4/20/2017 | Twilight[a] (18:20-18:30) | 24.9 ± 0.68 | 84.3 ± 3.32 | 56.1 ± 26.8 |
| | Start flashing/flying (18:30–18:40) | 23.8 ± 0.12 | 90.7 ± 0.76 | 10.5 ± 4.55 |
| 4/29/2017 | Twilight (18:16–18:26) | 18.7 ± 0.31 | 81.0 ± 1.28 | 136.5 ± 66.2 |
| | Start flashing/flying (18:26–18:36) | 17.7 ± 0.21 | 85.2 ± 1.03 | 6.49 ± 3.74 |
| 5/1/2017 | Twilight (18:22–18:30)[b] | 23.2 ± 0.36 | 81.8 ± 1.83 | 68.1 ± 21.0 |
| | Start flashing/flying (18:30–18:40) | 22.1 ± 0.28 | 87.5 ± 1.46 | 19.4 ± 8.65 |
| 5/18/2017 | Twilight (18:31–18:41) | 22.7 ± 0.07 | 90.3 ± 0.40 | 41.7 ± 15.97 |
| | Start flashing/flying (18:41–18:51) | 22.5 ± 0.04 | 91.5 ± 0.35 | 12.6 ± 5.34 |
| **B. Nanzhuang, Miaoli:** | | | | |
| 4/28/2017 | Twilight (18:26–18:30)[b] | 20.3 ± 0.46 | 71.2 ± 1.85 | 122.5 ± 26.3 |
| | Start flashing/flying (18:30–18:40) | 19.0 ± 0.46 | 77.2 ± 2.52 | 28.1 ± 24.2 |
| 5/7/2017 | Twilight (18:30–18:40) | 23.2 ± 0.07 | 92.8 ± 0.44 | 40.52 ± 18.2 |
| | Start flashing/flying (18:40–18:50) | 23.0 ± 0.07 | 93.6 ± 0.20 | 8.69 ± 4.15 |
| 5/8/2017 | Twilight (18:27–18:37) | 22.7 ± 0.06 | 93.35 ± 0.18 | 35.7 ± 14.0 |
| | Start flashing/flying (18:37–18:47) | 22.5 ± 0.05 | 93.8 ± 0.12 | 9.21 ± 4.01 |

Notes.

[a] For comparison, the environmental light intensity of twilight was estimated with ten minutes before fireflies flashing or flying in habitats. The recording interval is 10 s per time ($n = 60$).

[b] Recording time postponed due to unexpected schedule in field trip.

*ficta* emitted the brightest flashes among the studied female species, measuring up to 5.8 lux (or 850 nW/cm$^2$). In addition, the maximum luminescent intensity emitted from the five kinds of adult males was 2.3–14 lux (332.1–2,048 nW/cm$^2$), which is 1.01–7.26-fold higher than that of conspecific females (1.9–5.8 lux or 282.2–850 nW/cm$^2$). Thus, this result clearly shows that the range of the environmental light intensity (6.49–28.1 lux) when fireflies begin to flash partially overlaps with the luminescent intensity of fireflies. In addition, during 18:00–19:30, the change in average environmental temperature and relative humidity were in a range of 17.1–25.0 °C and 71.2–95.8%, respectively.

**Molecular phylogeny of Lampyridae inferred by *COI* barcodes**

To reveal how bioluminescence evolved, it is important to compare the luminescence spectrum and molecular phylogeny. The *COI* barcodes of eight studied species (except *C. costipennis*) were successfully sequenced for phylogenetic analysis (Table 2). All are new *COI* barcodes of Taiwanese fireflies sequenced in this study. Their haplotype sequences were deposited in GenBank under accession numbers MT534191–MT534201, ON209457.

The NJ tree (Fig. S2) and ML tree (Fig. S3) indicate that the studied genera *Abscondita*, *Curtos*, *Aquatica*, and *Luciola* belong to Luciolinae, while the genus *Pyrocoelia* belongs to Lampyrinae, a monophyly supported by previous mitogenomic phylogeny (*Wang, Wu & Wang, 2021*). However, the short *COI* sequences showed incongruence grouping among subfamilies in the high-level phylogeny. For example, *Rhagophthalmus* (Rhagophthalmidae) was placed close to the Luciolinae with a low bootstrap value; *Stenocladius* did not form a

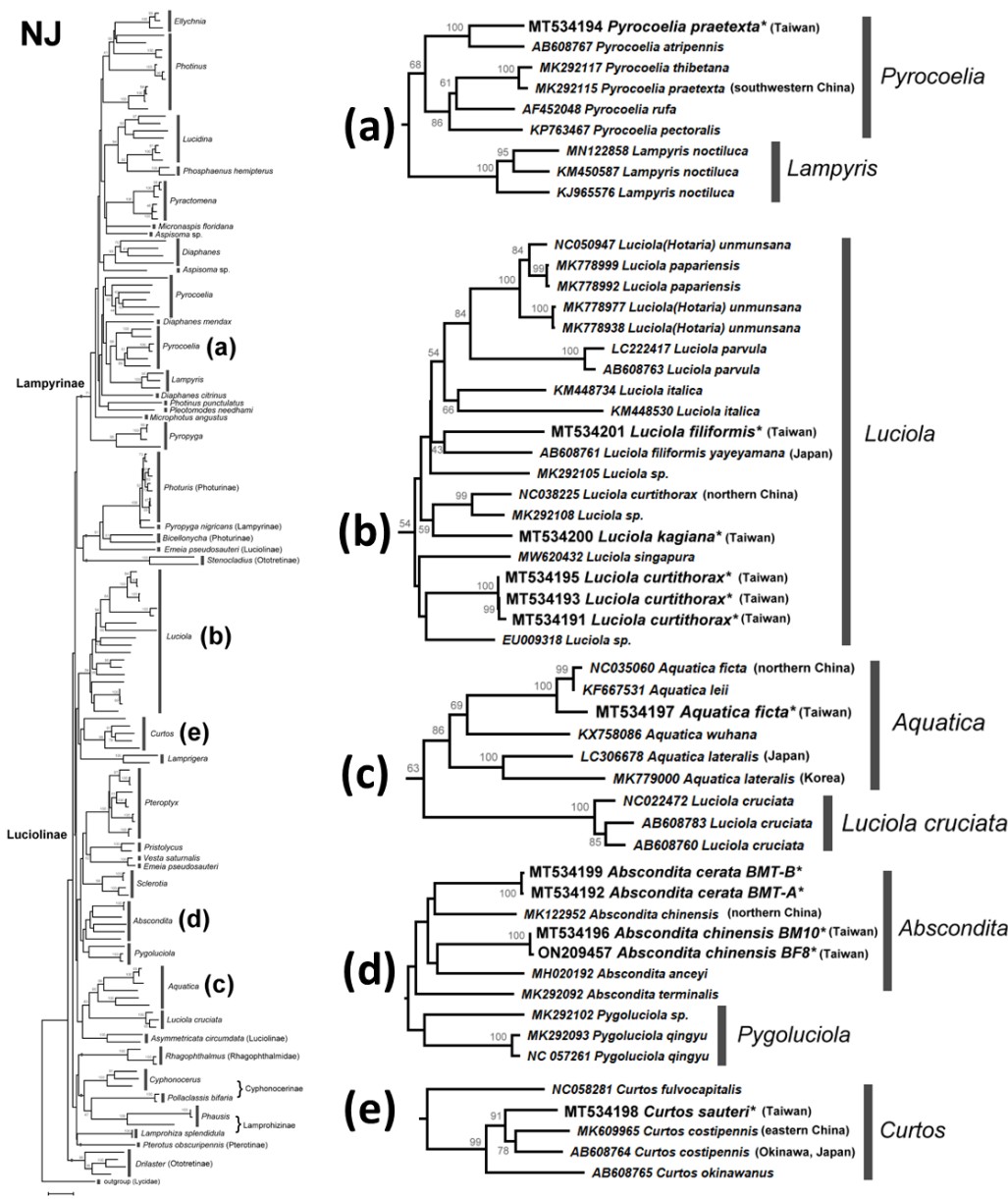

**Figure 4  Neighbor-Joining tree using the COI gene (520 bp) with bootstrap test results (500 replicates) at the nodes.** The optimal tree with the sum of branch length = 5.58552373 is shown. The evolutionary distances were computed using the Maximum Composite Likelihood method (*Tamura, Nei & Kumar, 2004*) with number of base substitutions per site. The rate variation among sites was modeled with gamma distribution (shape parameter = 1.079137891). All positions with less than 95% site coverage were eliminated. See Fig. S2 for a detailed NJ tree.

clade with *Drilaster* as Ototretinae. Nevertheless, most studied species are placed correctly with congeners (Figs. 4 and 5).

There are several monophyletic clades supported by medium or high bootstrap values. Lampyrinae was a monophyletic clade with *Pyrocoelia*, *Diaphanes*, *Lampyris*, *Microphotus*,

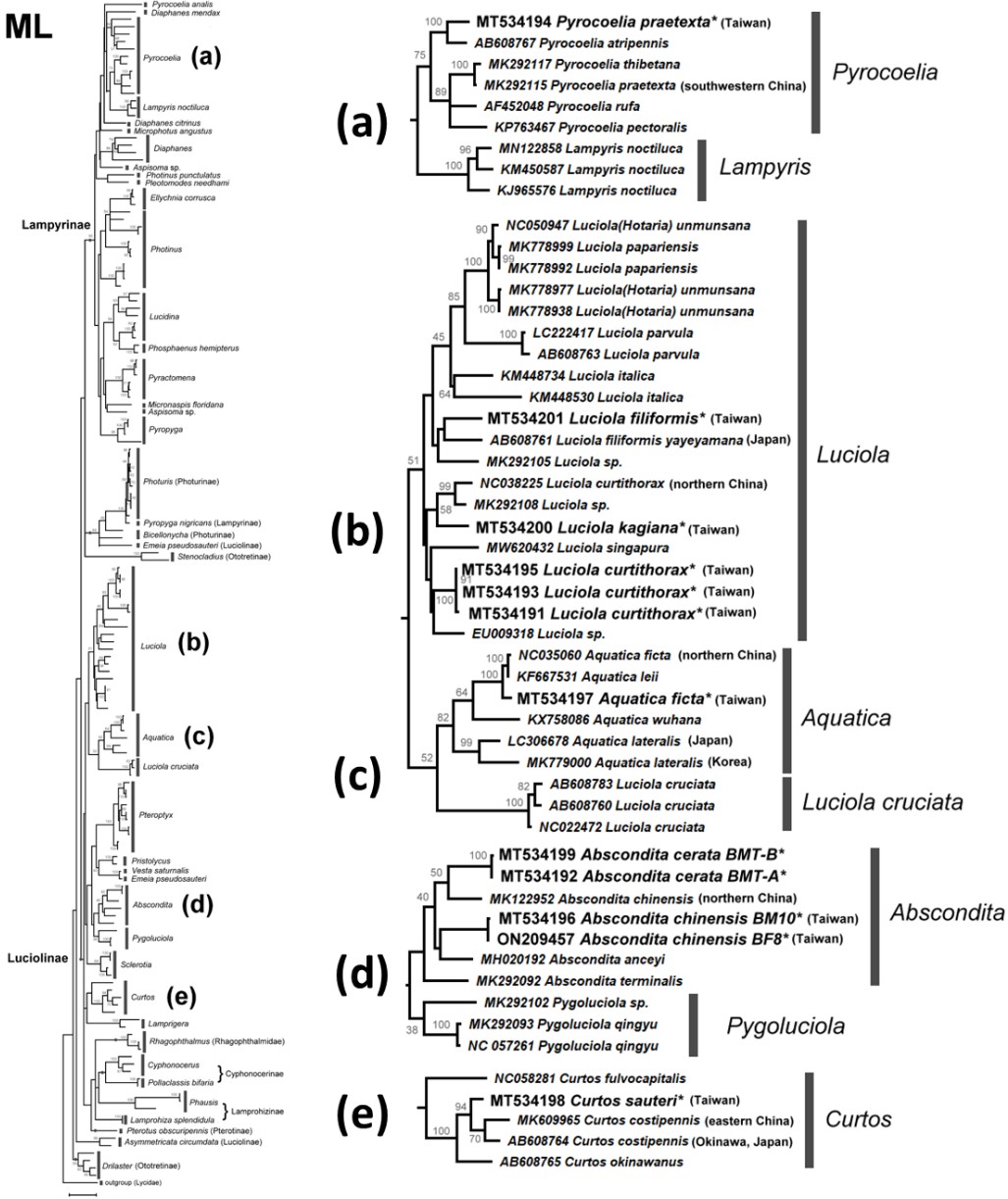

**Figure 5 Maximum Likelihood tree using the COI gene (520 bp) with bootstrap test results (500 replicates) at the nodes.** The evolutionary history was inferred using the Maximum Likelihood method based on the General Time Reversible model. The tree with the highest log likelihood (−11653.0821) is shown. Initial tree(s) for the heuristic search were obtained automatically by applying Neighbor-Join and BioNJ algorithms to a matrix of pairwise distances estimated using the Maximum Composite Likelihood (MCL) approach, and then selecting the topology with the superior log likelihood value. A discrete gamma distribution was used to model differences in evolutionary rates across sites (four categories (+G, parameter = 0.5737)). The rate variation model allowed some sites to be evolutionarily invariable ((+I), 37.4868% sites). The tree is drawn to scale, with branch length measurements based on the number of substitutions per site. All positions with less than 95% site coverage were eliminated. See Fig. S3 for a detailed ML tree.

*Aspisoma*, *Photinus*, *Pleotomodes*, *Ellychnia*, *Lucidina*, *Phosphaenus*, *Pyractomena*, and *Pyropyga*, although these genera did not form stable sister groups with each other. *Photuris* and *Bicellonycha* (as Photurinae) formed a clade with *Pyropyga nigricans*.

A previous study indicated that Luciolinae is not monophyletic, even when 436 gene loci were used (*Martin et al., 2019*). Thus, it is reasonable to see polyphyletic Luciolinae in the *COI* gene tree. The Luciolinae complex included the monophyletic genera *Luciola*, *Aquatica*, *Pterophyx*, *Sclerotia*, *Abscondita*, *Pygoluciola*, *Curtos,* and *Lamprigera*, which comprises *Pristolycus*, *Vesta,* and *Emeia pseudosauteri*. There are still several monophyletic clades with medium bootstrapping values. Excluding *L. cruciata*, the *COI* barcode grouped 12 *Luciola* species as a monophyly supported by a medium bootstrapping value (61/54), including the type species (*L. italica*). *Luciola cruciata* formed a stable clade with five *Aquatica* species. In addition, *Pygoluciola* clustered with *Abscondita* while *Curtos* clustered with *Lampyrigera* supported only with a low bootstrapping value.

Large *COI* sequence variations can be found between and/or within geographically distinct species. For example, two *COI* sequences of Chinese *Emeia pseudosauteri* were separated into distinct clades. *Emeia pseudosauteri* is restricted to central China and isolated among mountains. Such habitat isolation caused great mitochondrial DNA variation (*Liu & Fu, 2020*). Accordingly, there might be cryptic species and a need to reclassify some other species. A detailed analysis of Taiwanese fireflies will be discussed later.

## DISCUSSION

We identified five cohabitated species from Nankang and eight from Nanzhuang (Table S2). The evenings after sunny days with high humidity and cool temperature are the most suitable for firefly nocturnal activity (Table 6). Along with their morphological and genetic identification, we also measured the luminescence spectrum and luminescent intensity of firefly flashes, which might be related to the recognition of cohabitated fireflies. The biology of communication with flash patterns in fireflies is well outlined (*Lewis, Cratsley & Deiner, 2004*; *Stanger-Hall & Lloyd, 2015*). More than 10 cohabitated species can search for a conspecific mate at the same time *via* specific flash patterns (*Lloyd, 1969*). The males use this conspecific flash delay signaling for a particular female while females respond to male flashes with a species-specific response delay (*Lewis, Cratsley & Deiner, 2004*; *Lloyd, 1966*; *Lloyd, 1968*). A recent study (*Goh, Lee & Wang, 2022*) also recorded the species-specific flash patterns of three sympatric male fireflies (*Abs. cerata*, *L. kagiana*, and *L. curtithorax*). At least one previous study (*Ohba & Yang, 2003*) showed that the communication system of abundant *Abs. cerata* is classified as an HP system in which the female responses to the flying male flashes lasted about 0.24 s. Previous studies already revealed that flash patterns play an important role in conspecific fireflies' mating behavior. However, recording flash patterns in the field is not easy, especially when there is a short nocturnal activity period with a high population density of different cohabitated fireflies. This study further focuses on the flash color, luminescent intensity, and habitat environments to reveal other important factors that were previously lacking due to limited records of male–female communication signals. In addition, the COI phylogeny of the studied species revealed large genetic variation within known species in Taiwan and/or between adjacent regions.

## Differences in luminescence spectra among cohabitated fireflies

The $\lambda_{max}$ values of the luminescence spectra were similar within the same species, but different between species (Table 3). The pairwise comparison also showed significantly different interspecific $\lambda_{max}$, except between adult *Abs. chinensis* and adult *L. curtithorax* and between adult *C. sauteri* and larval *P. praetexta*. A recent study (*Goh, Lee & Wang, 2022*) recorded species-specific flash patterns of three sympatric male fireflies (*Abs. cerata*, *L. kagiana,* and *L. curtithorax*); both that study and the present study (Table 5) showed that males of *L. kagiana* and *L. curtithorax* have similar $\lambda_{max}$ values but still retain their own unique flash patterns. Thus, the four studied species with similar $\lambda_{max}$ might also have species-specific flash patterns, which need further study in the future. The above results might imply that most cohabitating fireflies distinguish between each other based on different luminescence spectra and/or specific flash patterns. This implication is important to consider when previous literature (*Chen, 2003*; *Chen & Jeng, 2012*) and our findings indicate that all studied adult fireflies appear simultaneously from April to June (Fig. 3). Thus, various cohabitated species may have evolved species-specific recognition to improve male–female searching within such a densely populated area over such a short nocturnal activity time.

Most fireflies have significantly different $\lambda_{max}$ between interspecific females (Table 4). Only those of *Aq. ficta* and terrestrial *Abs. cerata* females were not significantly different. However, the microhabitat of *Aq. ficta* and *Abs. cerata* was in the aquatic habitat and moist forest, respectively (*Chen, 2003*; *Jeng, Yang & Lai, 2003*). The flying males could still have better chance to find the conspecific females in their specific microhabitat.

In contrast, similar $\lambda_{max}$ were found in males of *L. curtithorax*, *Abs. chinensis,* and *L. kagiana* (Table 5), but males of *L. curtithorax* and *L. kagiana* have their own flash patterns (*Goh, Lee & Wang, 2022*). Thus, flash pattern is another key for cohabited female fireflies to recognize conspecific males (*Lewis & Cratsley, 2008*; *Lower, Stanger-Hall & Hall, 2018*). In addition, *Abs. cerata* and *L. kagiana* have different nocturnal activity time, while *L. curtithorax* is restricted to the dark ground layer of forest (*Goh, Lee & Wang, 2022*).

Based on the above phenomena, different $\lambda_{max}$, species-specific flash patterns, microhabitat choices, nocturnal activity time, and/or isolated mating seasons are key factors that may lead to the species-specific courtship of cohabited fireflies.

## Luminescent intensity of flashes implies sensing distance

Fireflies seem to be very sensitive to the photic environment in the evening. Artificial light pollution is a major force influencing firefly proliferation, mating, and growth (*Costin & Boulton, 2016*; *Firebaugh & Haynes, 2016*; *Haynes & Firebaugh, 2019*; *Owens, Meyer-Rochow & Yang, 2018*). The environmental light intensity and the light sensitivity of the fireflies influence whether the fireflies will flash. Therefore, the luminescent intensity of flashes emitted by fireflies could be an ecological indicator for evaluating light pollution to fireflies. This study further investigated this issue based on the first flash time of abundant *Abs. cerata* (Table 6). Fireflies start flashing or flying (nocturnal activity) when the environmental light intensity decreases to 6.49–28.1 lux ($\sim$950–4,114 nW/cm$^2$). The luminescent intensity of male *Abs. cerata* ranges from the average (2.1–3.4 lux or 406.6

± 96.5 nW/cm$^2$) to the maximum (14 lux or 2,048 nW/cm$^2$), which overlaps with the environmental light intensity suitable for their nocturnal activity (Table 1). A previous study also showed that the abundant *Abs. cerata* begins flashing when the photic environment decreases to 0.04–1.38 lux (*Ohba & Yang, 2003*). All imply that *Abs. cerata* could tolerate environmental light intensity around 28.1 lux but wait until 6.49 lux to start nocturnal activity in the evening at twilight. Another study also revealed that most male *Abs. cerata* start to fly in the evening at twilight while *L. kagiana* starts its nocturnal activity later (*Goh, Lee & Wang, 2022*), which Table 1 indeed showed lower luminescent intensity of *L. kagiana*. In addition, another study (*Owens, Meyer-Rochow & Yang, 2018*) revealed that half of the *Aq. ficta* specimens stopped flashing under bright exposure (∼20 and 200 lux). Table 1 further shows that the luminescent intensity of male *Aq. ficta* ranged from the average (3.1–4.1 lux or 525.7 ± 71.1 nW/cm$^2$) to the maximum (7.5 lux or 1102 nW/cm$^2$), which we also observed the *Aq. ficta* appeared with *Abs. cerata* during the same period of nocturnal activity in Nanzhuang. Such differences in luminescent intensity of the three species might imply another adaptation factor for the different nocturnal activity time among species.

Next, we measured the putative sensing distance between males and females. During a typical courtship, the flying males flash to attract perched females. Then, the female responds and flashes to the flying male. The male fireflies close and lands near the female; each displays different flash patterns for communication. As they court each other, the paired fireflies stop flashing on perch. Communication between female and male fireflies relies on the illumination of their light organ in the dark. Usually, the average luminescent intensity emitted by most females (the light organ from single tagma) is around half that of males (the light organ from double tagmata). The differences in luminescent intensity between sex could be due to their courtship behavior for sensing each other. The male needs a higher intensity exposure for females to find him while the female needs to save energy for later proliferation and only responds to male signals with detectable intensity.

The sensing distance between a female and male could be relative to their bioluminescent intensity. So, using the luminescent intensities of male and female, we could estimate the sensing distance. The assumption is the females have higher sensitivity while males have higher luminescent intensity. So, the luminescent intensity difference between male and female could be the sensing ability for a female to detect a male or vice versa. Thus, the maximum luminescent intensity might represent the maximum sensing distance between females and males, assuming that the minimum sensing distance (r, meter) is around the same luminescent intensity between females and males.

We can estimate the sensing distance using the example of the *Abs. cerata*. The males have a maximum luminescent intensity of 2,048 nW/cm$^2$ (14 lux) and the females have a maximum luminescent intensity of 282.2 nW/cm$^2$ (∼1.93 lux). Using the formula 14 / (r$^2$) = 1.93, we can estimate the maximum sensing distance (r) for this species to be around 2.7 m. Using the same formula calculation with average luminescent intensities, we estimated the average sensing distance to be around 1.8 m. In other words, the putative sensing distance for female *Abs. cerata* could range from 1.8 to 2.7 m, which may also be the sensing distance for a flying male searching for a female. That said, it is important to

note that most females prefer to perch as males fly to approach them (*Goh, Lee & Wang, 2022*), since perched females should flash less than what we measured. Thus, the sensing distance between males and females may actually be shorter. Nevertheless, the luminescent intensity could be an indicator of the sensing distance between flying males and perched females. After all, a previous study revealed that male *Photinus carolinus* use a 15–30 cm landing distance when approaching perched females (*Copeland, Moiseff & Faust, 2008*), which is a reasonable sensing distance in our estimation. Further behavior experiments should investigate these issues.

## Monophyly of *Luciola sensu stricto*

Both the mitogenome (*Jusoh et al., 2021*) and *COI* barcode (this study) revealed that each of the studied *L.* species form a clade, except for *L. cruciata*. *Luciola cruciata* and genus *Aquatica* were grouped together. The other genera of Luciolinae (*Curtos*, *Pteroptyx*, *Sclerotia*, *Abscondita*, *Pygoluciola*) are distinct.

## *Lamprigera* is not within Lampyrinae

Both the mitogenome (*Wang, Wu & Wang, 2021*) and 436 nuclear loci (*Martin et al., 2019*) indicated genus *Lamprigera* groups within Luciolinae instead of Lampyrinae. The *COI* phylogeny (Figs. 4 and 5) also showed that *Lamprigera* is a sister group to *Curtos* and separate from Lampyrinae. In addition, the morphology and *COI* sequences of eight native species (*Dong et al., 2021*) further revealed that *Lamprigera* should be closer to Luciolinae.

## Cryptic species implied by mitochondrial COI barcode variation

The mitochondrial genetic variation of fireflies within a population or adjacent regions has fewer genetic differences—*e.g.*, the desert-based *Microphotus octarthrus* (*Usener & Cognato, 2005*), the widespread *Photinus pyralis* (*Lower, Stanger-Hall & Hall, 2018*), and the Korean *Aquatica lateralis* (*Kim et al., 2001*; *Suzuki et al., 2004*). Previous biogeographical study revealed that the two studied sites in Northern Taiwan are within the same geographical regions; thus, we sequenced 1–3 individuals, except the abundant *Abs. cerata*, in which only one SNP site between two haplotypes could be found from nine *Abs. cerata* individuals of two habitats. Herein, the *COI* barcode showed a genus-level resolution for species identification in Figs. 4 and 5, although *COI* phylogenies in higher-level topologies are not consistent with those of previous morphological studies (*Ballantyne et al., 2013*; *Ballantyne et al., 2015*; *Ballantyne & Lambkin, 2013*; *Martin et al., 2017*; *Stanger-Hall, Lloyd & Hillis, 2007*) and molecular phylogeny (*Chen et al., 2019*; *Martin et al., 2017*; *Martin et al., 2019*; *Wang, Wu & Wang, 2021*). Nevertheless, the *COI* barcode could successfully identify most species at the genera-to-species level (Figs. 4 and 5). The *COI* phylogeny showed that the studied genera *Abscondita*, *Curtos*, *Aquatica,* and *Luciola* belong to Luciolinae, while *Pyrocoelia* belongs to Lampyrinae as expected.

The *COI* sequence variations revealed several cryptic species in Taiwan. For example, 62 SNP sites (~11.9% variation) were found in the *COI* sequences between Taiwanese and northern Chinese *Abs. chinensis*. Sixty-four SNP sites (~12.3% variation) were found in the *COI* sequences between *Abs. terminalis* and Chinese (Taiwanese) *Abs. chinensis*, respectively. Building on a previous study (*Ballantyne et al., 2013*), this study further

showed that the $\lambda_{max}$ of the *Abs. chinensis* lantern spectrum (flash color) is different between Taiwanese (572 nm) and northern Chinese (565 nm) individuals, though there might be unknown environmental effects that cause the flash color variation in widespread species, like with the North American firefly, *Photinus pyralis* (*Lower, Stanger-Hall & Hall, 2018*). Thus, the new evidence reveals that Taiwanese *Abs. chinensis* may be a distinct species to Chinese *Abs. chinensis* (Figs. 4D and 5D).

Large *COI* variation was also found in six Asian species: *P. praetexta* (Figs. 4A and 5A), *C. costipennis* (Figs. 4E and 5E), *Aq. ficta* and *Aq. lateralis* (Figs. 4C and 5C), and *L. curtithorax* and *L. filiformis* (Figs. 4B and 5B). The *COI* barcode also indicated 17 SNP sites (~3.27% variation) between Taiwanese and northern Chinese *Aq. ficta* (Figs. 4C and 5C). One study indicated that the characterization of the Chinese *Aq. ficta* differed slightly from the Taiwanese *Aq. ficta* (*Ballantyne & Lambkin, 2009*). In addition, *Aq. leii* was considered as a different species to the Chinese *Aq. ficta* (*Fu, Ballantyne & Lambkin, 2010*). However, there are only two SNP sites (~0.39% variation) between Chinese *Aq. ficta* and *Aq. leii*. The Taiwanese *Aq. ficta* may be a cryptic species with a large variation (17 SNP sites). In contrast, the *COI* barcode indicated a large variation of 10% (52 SNP sites) between Korean and Japanese *Aq. lateralis* (Figs. 4C and 5C). There are 58 SNP sites (~11.2% variation) between Taiwanese and northern Chinese *L. curtithorax* (Figs. 4B and 5B). The *COI* barcode indicated 58 SNP sites (~11.2% variation) between Taiwanese *L. filiformis* and Japanese *L. filiformis yayeyamana* (Figs. 4B and 5B). The *COI* barcode indicated 57 SNP sites (~11% variation) between Taiwanese and southwestern Chinese *P. praetexta* (Figs. 4A and 5A). The *COI* barcode indicated 33 SNP sites (~6.35% variation) between southern Japan and eastern Chinese *C. costipennis* (AB608764 and MK609965 in Figs. 4E and 5E). All these examples indicate large *COI* variations between two geographical isolates. Further investigations are needed to reclassify these geographically isolated species.

## Bioluminescent evolution inferred from mitochondrial COI barcodes and known phylogeny of Lampyridae

The contracted high-level phylogeny (*Chen et al., 2019*; *Martin et al., 2019*) and Luciolinae grouping (*Jusoh et al., 2021*) correspond well with our bioluminescent evolution phylogeny (Fig. 6). Another study revealed the bioluminescent evolution *via* recombinant luciferases and suggested the origin of beetle bioluminescence (*Oba et al., 2020*). Accordingly, this study gives a detailed summary on the evolution of bioluminescence in Lampyridae based on the $\lambda_{max}$ of its luminescence spectrum (*Arnoldi, Neto & Viviani, 2010*; *Goh, Lee & Wang, 2022*; *He et al., 2021*; *Oba et al., 2020*; *Wilcox, 2021*). Our studied species further revealed that the fireflies' luminescence color was originally a green color in a Lampyridae ancestor, then red-shifted to a yellow-green in Luciolinae and is now an orange-yellow color in some derived species (Fig. 6).

## CONCLUSION

This study establishes the bioluminescent spectrum and intensity of nine cohabitated fireflies and can be referenced to ensure that light pollution in habitats does not become high enough to disrupt firefly mating. The mitochondrial COI barcode revealed a genus-level

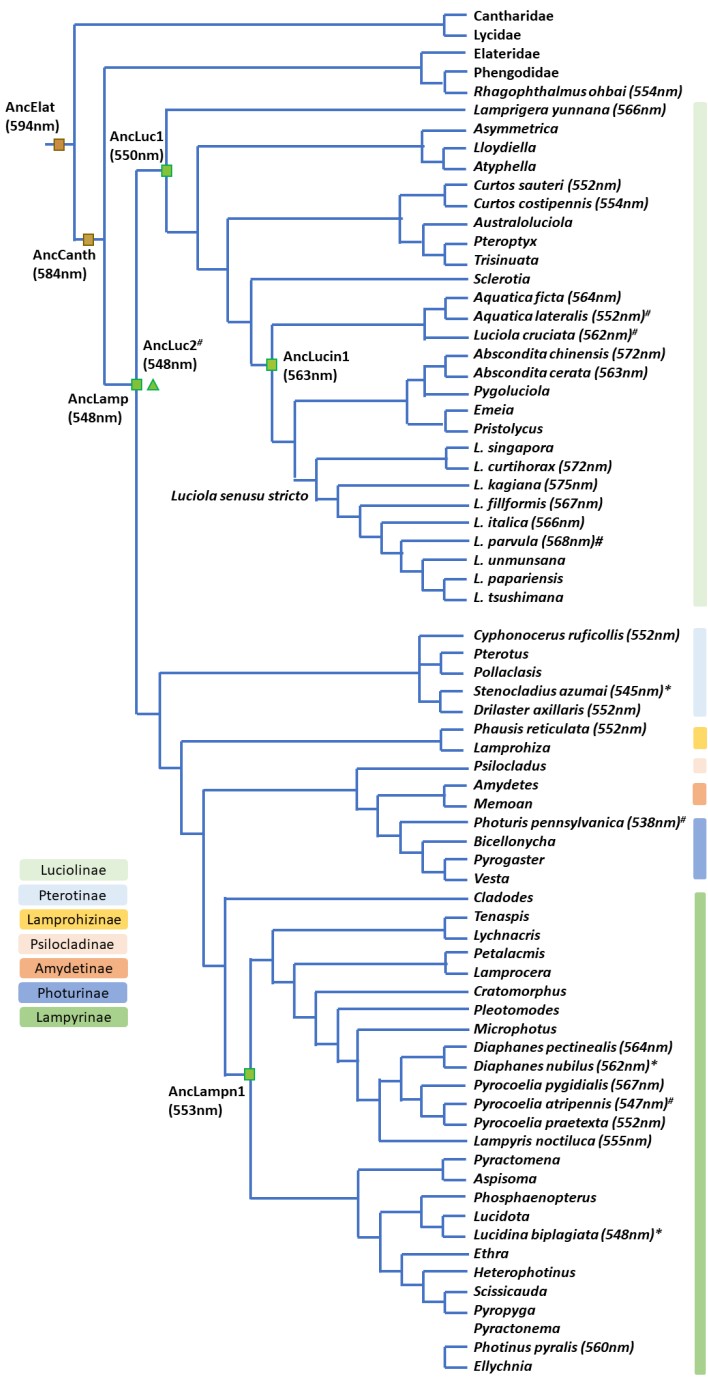

**Figure 6** **Bioluminescent evolution of fireflies.** The phylogenetic topology and lantern wavelength ($\lambda_{max}$) were adopted from our new data and previous studies (*Chen et al., 2019*; *He et al., 2021*; *Jusoh et al., 2021*; *Martin et al., 2019*; *Oba et al., 2020*).

resolution for species identification and six cryptic species that need to be further studied. Combined with previous literature, this study supports the argument that bioluminescent evolution has red-shifted to yellow-green in Luciolinae and specified to orange-yellow color in some derived species.

## ACKNOWLEDGEMENTS

We are grateful to Miss Xian-Ju Chang for her sampling assistance. Thanks also to Noah Last of Third Draft Editing for his English language editing. We extend our deepest thanks to reviewer Christine Lambkin, Queensland Museum, for her valuable comments.

### Funding

This research was funded by Academia Sinica, Taiwan. The funders had no role in study design, data collection and analysis, decision to publish, or preparation of the manuscript.

### Grant Disclosures

The following grant information was disclosed by the authors:
Academia Sinica, Taiwan.

### Competing Interests

The authors declare there are no competing interests.

### Author Contributions

- King-Siang Goh conceived and designed the experiments, performed the experiments, analyzed the data, prepared figures and/or tables, authored or reviewed drafts of the article, and approved the final draft.
- Liang-Jong Wang conceived and designed the experiments, performed the experiments, analyzed the data, prepared figures and/or tables, authored or reviewed drafts of the article, and approved the final draft.
- Jing-Han Ni performed the experiments, analyzed the data, prepared figures and/or tables, and approved the final draft.
- Tzi-Yuan Wang conceived and designed the experiments, performed the experiments, analyzed the data, prepared figures and/or tables, authored or reviewed drafts of the article, and approved the final draft.

### Data Availability

The COI sequences are accessible at GenBank: MT534191–MT534201, ON209457.

### Supplemental Information

Supplemental information for this article can be found online at http://dx.doi.org/10.7717/peerj.14195#supplemental-information.

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
