# Peer review of "Luminescent characteristics and mitochondrial COI barcodes of nine cohabitated Taiwanese fireflies"

_PeerJ, doi:10.7717/peerj.14195_

## Round 0.1 · original submission · Major Revisions

Dear Dr. Goh and colleagues:

Thanks for submitting your manuscript to PeerJ. I have now received two independent reviews of your work, and as you will see, one reviewer recommended rejection, while another suggested a major revision (with many suggested changes). I am affording you the option of revising your manuscript according to both reviews but understand that your resubmission may be sent to at least one new reviewer for a fresh assessment (unless the reviewer recommending rejection is willing to re-review).

The reviewers raised many concerns about the manuscript. Please address all of these in your rebuttal letter. I especially would like to see your response to insufficient literature being referenced. Overall, there is a lack of completeness with your work and all sections of the manuscript appear to need revision. The Discussion in particular fails to frame your findings within the existing knowledgebase of the field.

There are many minor suggestions to improve the manuscript. Note that reviewer 2 kindly provided a marked-up version of your manuscript.

Therefore, I am recommending that you revise your manuscript, accordingly, taking into account all of the issues raised by the reviewers. If your next revision does not address the concerns of the reviewers your work will be rejected.

Good luck with your revision,

-joe

Reviewer 1 ·

Basic reporting

The introduction needs a bit more work. Background for this study is not clearly stated, and the objectives are confusing. The authors could have discussed more compelling issues. They can discuss the latest advancement of DNA barcodes in fireflies (there have been many studies), sympatric species in fireflies or review previous studies on the impact of artificial light pollution at night (ALAN) on fireflies.

Previous studies are referenced, although some are not relevant, e.g. firefly algorithm. Perhaps the authors can provide a bit more background about previous work on fireflies in Taiwan.

Experimental design

Research questions are not well defined with some are irrelevant to the main objective (or not clearly stated). The research’s knowledge gap is unclear.

Methods described with sufficient detail, but because the objectives are not well defined, some methods are irrelevant.

Validity of the findings

Presentation of data is confusing because data are not linked to research questions

DNA barcode seems to be the main focus but result and discussion are not well presented.

Conclusions are not well stated and unfittingly linked to the study.

Additional comments

This study offers exciting findings on the use of DNA barcodes, the light spectrum and comparative morphology of fireflies in Taiwan. I commend the authors for their extensive work in the field and lab. However, the current form of the manuscript is poorly written and their report of findings can be very confusing.

·

Basic reporting

see attached file for explanation and more details:
Is the paper correct with respect to the issues and literature involved? No, inadequate background information especially biology, and insufficient comparison with previous work to place the importance of this work into a scientific context.
Many references to previous work not included.

Are the interpretations and conclusions justified and adequate? No. Discussions are often a repeat of results with, few and sometimes inaccurate comparisons to other studies, no acknowledgement of limitations and very little discussion. Discussions should have compared results with what found before, and suggested hypotheses to explain results especially where they differed from previous studies, and placed their study of Taiwanese fireflies into a world/Asian context. Conclusions are poorly explained, not based on results, and sometimes irrelevant.
Conclusions should have summarised their discussion, highlighting significant findings.

Experimental design

see attached file for explanation and more details:
Is the paper a new contribution to the topic? YES, and unfortunately the ‘new’ work is not highlighted.

Are the methods adequate and sufficiently well described? Adequate just, more details would be appreciated though. Methods should have used methods previously used or explained how they differed and why

Have the appropriate statistical analyses been performed? Not well explained

Is the information presented in a logical sequence and concise manner? Not completely, some sections need to be moved (as noted on MS), and in areas too concise. Background/introduction should have outlined what was already known for – gender differences, population differences, species differences for - measurements; light intensity, light spectra, COI

This paper needs (possibly in the introduction, if not, in the relevant parts of the discussion) an outline of the biology of fireflies that links the disparate studies included
1- light spectra, light intensity, mating behaviour; mate, species, and generic recognition
2 - morphology, DNA, relationships

Validity of the findings

see attached file for explanation and more details:
Considerable work has been done, is new, and worthy of publication.

However, the paper shows little understanding of the biology of the group, has not referenced many similar studies, has no discussion, few and inaccurate comparisons to previous studies, and the most unusual conclusions! I have spent over a week on this, and have had enough, and have now stopped.

As it is currently, the paper is insubstantial because:
• description of methods are minimal require more detail
• discussion and conclusions are inadequate as they are currently a summary of results
• insufficient and unsatisfactory discussion based on evidence from results
• no acknowledgement of weakness of data or results that compromise ability to make well-supported inferences
• few, inaccurate, and incomplete comparisons of results with previous scientific work:

Additional comments

see attached files for explanation and many more details.

---

## Round 0.2 · accepted · Accept

Dear Dr. Goh and colleagues:

Thanks for revising your manuscript based on the concerns that were raised. I now believe that your manuscript is suitable for publication. Congratulations! I look forward to seeing this work in print, and I anticipate it being an important resource for groups studying firefly biology and ecology. Thanks again for choosing PeerJ to publish such important work.

Best,

-joe